# Preparation, Characterization of New Antimicrobial Antitumor Hybrid Semi-Organic Single Crystals of Proline Amino Acid Doped by Silver Nanoparticles

**DOI:** 10.3390/biomedicines11020360

**Published:** 2023-01-26

**Authors:** R. S. Almufarij, A. E. Ali, M. E. Elbah, N. S. Elmaghraby, M. A. Khashaba, H. Abdel-Hamid, H. A. Fetouh

**Affiliations:** 1Department of Chemistry, College of Science, Princess Nourah bint Abdulrahman University, P.O. Box 84428, Riyadh 11671, Saudi Arabia; 2Chemistry Department, Faculty of Science, Damanhour University, Damanhour 22511, Egypt; 3Chemistry Department, Faculty of Science, Alexandria University, Alexandria 21321, Egypt

**Keywords:** silver nanoparticles, proline, cobalt chloride, single crystal, antitumor, anisotropy

## Abstract

Proline is water soluble amino acid extensively used in drug delivery systems. Compounds of cobalt (Co) transition metal have potent antimicrobial and anticancer activities. However, a drug delivery system combining proline cobalt is not reported yet. For the first time, new hybrid semi-organic single crystals of proline cobalt chloride (PCC) are prepared. The novelty of the article is also that single crystal proline cobalt chloride showed potent antimicrobial and antitumor activity. Doping of PCC by Ag^0^NPs significantly increased these biological activities. The anisotropic magnetic properties of single crystals can mitigate the cytotoxicity of Ag^0^NPs on normal cells. Silver nanoparticles (Ag^0^NPs) improved the crystal habits and physicochemical properties. Ag^0^NPs showed the best performance, paramagnetic materials n-type semiconductors due to delocalized excess electrons of Ag^0^NPs incorporated in the crystal lattice interstitially. Crystals have high absorptivity for UV-radiation electromagnetic radiation. Ag^0^NPs enhanced AC electrical conductivity up to 2.3 × 10^4^ Ω cm^−1^ due to high electron density. Proline doped crystals are obtained in good purity as triclinic unit cell with having anisotropic magnetism. PCCAg^0^NPs crystal exhibited: high antimicrobial activities to various bacterial and fungal species, inhibition zone (mm): 21, 25, 24, 26, 30, 28, 12, and 46 for *S. aureus*, *E. faecalis*, *S. typhi*, *E. coli*, *P. aerugino*, *K. pneumoniae*, *A. braselienses*, and *C. albicans*, respectively, in comparison to ciprofloxacin antibiotic (23, 0, 26, 26, 25, 0, 0, 0) for the same tested species, respectively; higher cytotoxicity against breast cancer cells (IC_50_ 22.1 μM) than the reference drug cisplatin (IC_50_ 11.7 μM); and lower cytotoxicity to normal healthy lung cells MRC-5, (IC_50_ 145.5 μM) than cisplatin (IC_50_ 30.2 μM). Hence, this crystal is a candidate for chemotherapy of breast cancer.

## 1. Introduction

Single crystals in literature are described as polar crystals, thermoelectric materials that generate electricity on heating that change intrinsic magnetic moments and permittivity; tiny small sensors in electronic and power generators convert chaotic waste heat into useful electric work [1,2]. They store electrical energy after removal of an applied electric field as charge-reservoirs parallel plate capacitors; piezoelectric materials applied in automobiles electronics, touch screens of laptop and mobile phones; microwave filters, and energy storage systems [3,4]. In electric fields, the electron clouds in atoms polarize. The dielectric constant or refractive index is affected by the frequency of absorbed quantized energy UV-Vis.; and absorb UV-Vis. of sunlight generating electron–hole pairs [5]. The current flow depends on photon energy. Polarization in electric fields is controlled by dielectric constant; and AC conductivity measures macroscopic dielectric properties and polarization [5]. Dielectric materials are used as capacitors, insulators and semiconductors. Ferroelectric dielectric perovskite crystals undergo a phase change from a non-polar para-electric phase to a polar ferro-electric polar phase depending on heating or pressure at Curie temperature, *T*_C_ providing no thermal decomposition at *T*_C_, and are applied as electrostrictive actuators due to strong electronic power generation and saving [6,7].

Single crystals of glycine amino acid have high nonlinear optical (NLO) and ferroelectric properties [8]. Glycine cobalt chloride (GCC) single crystals are used in: infrared detectors [9]; and pervoskite crystals used in conversion waste heat of combustion, exhausted from pipes, automobiles, batteries, furnaces, and chimneys into electricity [10].

No details are reported about doping glycine and proline-single crystals by Ag^0^NPs [11]. Perovskites metal oxides single crystals have a molecular formula, ABO_3_ are pillars in electronics and nanotechnology as superconductive electrodes, magnets, insulators, etc.; have cubic or semi-cubic structure; and applied ferroelectric piezoelectric lead-based perovskites. Examples include Pb(Zr_x_Ti_1−x_)O_3_ or Pb(Mg_0.33_Nb_0.67_)O_3_-PbTiO_3_, which are low cost; however, lead is a toxic element [12]. The phases bithmus titanate Bi_4_Ti_3_O_12_ and barium titanate BaTiO_3_, alkali Li, Na, K niobates NbO_3_, bismuth-alkali titanates (Na_0.5_Bi_0.5_)TiO_3_, K_0.5_Bi_0.5_TiO_3_ and their solid solutions (Ba, Ca)(Ti, Zr)O_3_, (Na_0.5_Bi_0.5_)TiO_3_-BaTiO_3_ are ecofriendly crystals [13,14].

Hybrid semi-organic single crystals (HSOSCs) have ferroelectric and magnetic properties in a single phase and are good alternatives to conventional materials such as BiFeO_3_, BiMnO_3_, FeO_3_, and LaFeO_3_. Polar semi-organic crystals possess ferroelectricity, antiferroelectricity, piezoelectricity, thermal stability, metallic conductivity, superconductivity, ferromagnetism, anti-ferromagnetism, etc. [15,16].

The use of Ag^0^NPs as bridging linkers in proline single crystals is not reported. This study aims to prepare, characterize and enhance the quality and physicochemical properties of new single crystals (structure, magnetic properties and NLO activity by Ag^0^NPs doping). The synergistic effect of dopants CoCl_2_ and Ag^0^NPs could affect such properties.

## 2. Experimental

### 2.1. Materials and Methods for Preparation of Single Crystals

All chemicals used in this study are all of analytical grades used as received without further purification. Molecular weight and purity of glycine, proline, silver nanoparticles (Ag^0^NPs) and cobalt chloride hexa-hydrate, CoCl_2_·6H_2_O, are collected in Table 1.

Appropriate salts weights are mixed in stoichiometric molar ratio: (glycine or proline)_1−x_ (CoCl_2_)_x_ (x is dopant weight percent of either CoCl_2_ in absence and presence of Ag^0^NPs). Glycine single crystal is grown as a control single crystal.

Single crystals are grown following a slow evaporation method [12], at optimum experimental conditions: temperature 25 °C ± 0.1, pH 5.5 and 100 rpm agitation speed for 2 h. The salts mixture is continuously agitated in deionized water green nontoxic solvent until attaining a homogeneous saturated solution at the same temperature. The saturated solution is filtered and left covered with porous filter paper. Crystal nucleation and growth are allowed through slow water evaporation. Pink colored single crystals: proline cobalt chloride (PCC) in the presence of 0.15 wt.% Ag^0^NPs are harvested after three weeks of representative visual appearances as shown in Figure 1 in comparison to glycine cobalt chloride (GCC).

High quality crystals have a large size, and perfect octahedral (Oh) geometry with defined edges. Crystals containing Ag^0^NPs have a more intense pink color, suggesting applications as new colored materials in the second harmonic generation.

### 2.2. Characterization of Single Crystals

Crystals are characterized using different spectroscopic methods of analysis. Carbon, hydrogen, and nitrogen CHN elemental analysis (EA) is determined using Malvern analytical elemental analyzers. The content of Co(II) ion is determined via several digestion decompositions in aqua-regia to dissolve organic matter. Cobalt residue is dissolved in double distilled water and is determined using a Shimadzu 6650 atomic absorption spectrophotometer.

Fourier transformer infrared (FTIR) spectra are recorded using Bruker Tensor 27FTIR-spectrophotometer at frequency range 400–5000 cm^−1^, and Nujol mull UV-Vis. electronic spectra are recorded using Lambda 4B Perkin Elmer spectrophotometer at wavelength range 200–900 nm. Molar magnetic susceptibilities and Pascal’s constants are determined using Faraday’s method at 25 °C while calibration spectrophotometer using Hg[Co (SCN)_4_].

Thermogravimetric analysis, TGA and differential thermal analysis, DTA are determined using Shimadzu DTA/TGA-50, heating rate 10 °C/min, platinum cell under nitrogen, and flow rate 20 mL min^−1^ [17,18].

Powder X-ray diffraction at 2θ range 5–80° with Cu-Kα X-ray (λ 1.54 Å) radiation source. Density, ρ, is determined by floatation technique in a saturated solution of NaCl, KBr and benzene separately. The number of formula units per unit cell (Z) is calculated by using the equation [17,18]:(1)$$Z=\frac{\rho \mathrm{NV}}{\mathrm{Mw}.}$$
where V is volume of unit cell, and N is Avogadro’s number.

X-band electron spin resonance spectra at room temperature using a reflection (JES-RE1X ESR. ESR spectrometer) at 9.43 GHz in cylindrical resonance cavity, 100 kHz modulation, 5 mW electric power and LMR Gauss meter control applied magnetic field. 

AC electrical conductivity of single crystal sample is measured using four probes Agilent 4294 A Impedance Bridge applying sine AC signal, 10 amplitude. Thin gold layers (10 nm) are deposited on two opposite sides of the pellet sample by thermal evaporation under vacuum 10^−5^ mbar using Joule evaporator. Silver wire is glued on each deposit with silver lacquer.

Antimicrobial activity is determined using paper disk diffusion method against Gram-positive bacteria: *Staphylococcus aureus*, *Enterococcus faecalis*; Gram-negative bacteria: *Escherichia coli*; *Pseudomonas aeruginosa*, *Salmonella typhi*, *Klebsiella pneumoniae*); and Fungi: *Aspergillus brasiliensis*; *Candida albicans*. Cytotoxic activity this metal complexes on cancer cell lines are screened using MTT assay against MCF-7 (human breast adenocarcinoma), and healthy MRC-5 human lung fibroblasts (control cell lines). Results of in vitro cytotoxic activity are compared with reference standard Cis-platin in terms of IC_50_.

## 3. Results and Discussion

The chemical composition and atomic percent are collected in Table 2.

High atomic percent C, H, N, and O atoms indicated that proline and glycine amino acids are the mother materials for crystals [19]. Both proline and glycine have a white color. Doping of the crystal lattice of both proline and glycine by CoCl_2_ produced optically active have high molecular weight (Mw.) single crystals of pink color. Ag^0^NPs intensified the pink color of PCC and increased Mw. up to 1051.54 g mol^−1^ forming a self-supramolecular assembled single crystal [20]. CoCl_2_ incorporated into glycine and proline, forming a crystal with a 1:2 molar ratio. The crystals are stable, non-hygroscopic and soluble in water polar green solvent.

The infrared spectra of the crystals are compared with that of amino acids to deduce the intercalation mode. The charge transfer from ligand (proline or glycine) to Co(II) ion decreased the force constant of bond causing red shift of bond position and enhanced optical activity of crystals. Some blue shift occurs on back donation of the electron from Co(II) ion to the electron donor atom to reinforce the coordinate bond [21]. Assignments of main spectral vibrational band are given in Appendix A.

Proline bands at 3430, 2983, 2504 and 1312 cm^−1^ are assigned to υ(OH), υ_as_(CH_2_), υ_s_(CH_2_), and υ(C-N), respectively. υ, δ, and γ vibration of NH_2_^+^, υ_C-O_ of carboxylate COOH group vibrational bands at 3066, 1623, 871, and 1290 cm^−1^, respectively. Bending vibration bands of CH_2_ pyrrolidine ring of proline appeared at 1454, 1402, 1367, 1318, 1164, 834, 589 and 538 cm^−1^, respectively. The absence of a ν_COOH_ band at 1725 cm^−1^ is due to the deprotonated COOH group in Zwitter ion form. Two bands at 1559 cm^−1^ and 1357 cm^−1^ signified asymmetric and symmetric stretching of the deprotonated carboxylate COO^−^ group, respectively [22].

FTIR spectral bands of proline are compared with that of single crystal to declare bonding mode with Co(II) ion. IR spectral bands of two PCC_2_ crystals showed a ν_COOH_ band at the 1730–1732 cm^−1^ range (Carbonyl group). The absence υ_asy COO–_ is due to protonation on coordination to the Co(II) ion. υ_NH_ and δ_NH_ are red shift by 101–107 and 17–19 cm^−1^, respectively, relative to proline. υ_C-N_ and *γ*_NH_ changed in shapes and positions, indicating the participation of a N atom in chelation. PCC showed a new band at 437–441 cm^−1^ (υ_Co-N_). IR spectra explored amino acid is bidentate ligand coordinate Co(II) ion through N, O atoms, see Figure 2.

Optical activity confirmed electronic spectral and magnetic properties of crystals investigated by using Nujol mull absorption spectroscopy at room temperature [23,24]. Calculated optical parameters: ligand field splitting and stabilization energy (CFSE), 10 Dq are collected in Table 3. Ligand field parameters for Co(II)-Racah inter-electronic repulsion parameter *B*′: 595–753 cm^−1^. The lowering *B* of free Co(II)ion complexation suggests orbital overlap and electrons delocalization on Co(II)ion. In nephelauxetic ratio, the *β* less than one indicating partial covalent bond “σ” between Co and amino acid [25]. The parameters of tetragonal distortion in crystals (Ds and Dt) and the crystal field parameter (Dq) are derived from the energy of different electronic transitions.

The values of magnetic moment have B.M suggesting high spin distorted tetragonal geometry, (t_2g_)^5^(e_g_)^2^ configuration and ^4^A_2g_ ground state. PCC displayed five absorption bands indicating axial distorted Oh symmetry around Co(II) ion [26] due to transitions ν_2_ ^4^A_2g_ → ^4^B_2g_, ν_3_ ^4^A_2g_ → ^4^E_g_^(b)^ and ν_4_ ^4^A_2g_ → ^4^B_1g_, ^4^A_2g_ → ^4^E_g_^(c)^ [^4^T_1g_ (P)] and ^4^A_2g_ → ^4^A_2g_(c) [^4^T_1g_ (P)]. Bands corresponding to ν_1_ ^4^A_2g_ → ^4^E_g_^(a)^ were not observed in the spectra [27,28].

Thermal degradation confirmed the molecular structure and thermal stability. TGA and DTA thermograms of crystal are shown in Figure 3, Figure 4 and Figure 5. TGA showed weight loss of tested sample as a function of temperature or time. DTA measures temperature difference (ΔT = T_S_ − T_R_) between the sample (S) and reference (R) materials at zero heat flow difference (ΔH = H_S_ − H_R_ = 0).

TGA and DTA of proline and crystals showed distinguished coordination and stability ranges in peak temperatures and kinetic parameters. Shape index symmetry of peak “S” ratio of slopes of curve tangents at inflection points a/b depends on reaction order, n (1°, 2° order, etc.) and is determined from DTA, see Appendix A [29].

Applying least square method, the plot (ln ΔT versus 1/T) is represented in Appendix A and gave straight lines obeying Arrhenius relation. Activation energy E_a_ of decomposition is calculated [30]. The TGA %wt.loss for decomposition steps is correlated to the proposed chemical formula, see Table 4.

Proline showed 2.14% wt.loss at 199 °C, which caused a weak DTG peak at 66 °C corresponding to dehydration. Thermal degradation from 199–266 °C, 97.86% wt.loss due to complete decomposition is associated with broad DTG peak at 244.37 °C, in a narrow temperature range indicating rapid thermal decomposition. The corresponding Ea kJ/mol^−1^ and n values for two exothermic steps are 32.42, (1.05) and 12.88, (1.09), respectively [31]. Consecutive thermal degradation followed 1° order kinetic according to Figure 1.

PCC five degradation stages: 24.48% wt.loss is due to partial dehydration and removal six water molecules give strong DTG peaks, and *T_max_* 82 °C. Decomposition steps are at temperature ranges 192–302, 302–366 and 366–600 °C. In DTG weak, medium and strong peaks located at 214, 327 and 537 °C, respectively. Thermal decompositions steps suggested elimination: two coordinated H_2_O molecules, 2OH group and fraction residue 4 C atoms. The final degradation step showed wt.loss 13.02% at 600 °C, DTG peaks at 753 °C is due to release 1.0 mole H_2(g)_ + 2.0 moles NH_3_ (E_a_ 122.22 kJ/mol, n 1.41), respectively. The final residue is CoO + 6C. Thermal decomposition pathway is represented in Figure 2 [32].

TGA thermograms PCC and PCCAg^0^NPs showed nearly similar behavior indicating isostructural and isothermally behavior. PCCAg^0^NPs crystal exhibited five decomposition stages. wt.loss at 70.88–195.4°C is due to removal of lattice-water molecules with sharp strong DTG peaks at 95.38 °C at narrow temperature ranges signifying rapid thermolysis. Anhydrous crystal started decomposition through two overlapped continuous steps at 195.4–261.88 °C and 261.88–305.5 °C giving weak broad diffused DTG peaks at: 210, 276 and 320 °C, respectively, wt.loss 5.24 and 1.70% due to loss 4.0 coordinated H_2_O molecules. Fourth and fifth thermal decomposition processes Wt.loss 12.04 and 22.66% due to release 2H_2_O + C_7_H_6_ and C_13_H_14_ + 2NH_3(g)_, respectively, leaving AgCo_2_O_5_ + C residue. These steps showed small different thermal stability and T_max_. DTG peaks showed increasing ΔH due to Ag^0^NPs interaction with OH^−^ of proline forming a robust self-assembled monolayer on Ag^0^NPs via strong Ag O covalent bond and Van der Waals interaction. The bond energy Ag-O is 217 kJ mol^−1^. There is a small difference in ΔH confirmed supramolecular structure [33], see Figure 3.

By profile fitting and indexing the PXRD pattern, the crystal structure is solved using direct methods with simulated annealing implemented in software EXPO2014. The unit-cell parameters for PCC crystals are refined using Pawley/LeBail fit analysis [17,34]. PXRD data of PCCAg^0^NPs crystals are indexed with N-TREOR, and in Figure 6, the crystal lattice parameters are collected in Table 5 and Table 6 including unit cell parameters and Rietveld refinements Rp, Rwp, and S 2.03 parameters are also described.

Figure 7 showed the refined bond distance in PCC and PCCAg^0^NPs.

The crystal structure is shown in Figure 8.

Figure 9 demonstrated the refined pXRD of PCCAg^0^NPs.

Refinement PXRD patterns convert the approximate structure of PCCAg^0^NPs crystal into an actual structure, see Figure 10.

NH group and O atoms of proline give bidentate chelating, forming a distorted octahedral geometry because imperfect bond angles. The equatorial plane is by O3, O16 of 2 COOH groups and (N6, N14) atoms. Bond angles ranges 84.83 (2)–95.74 (5)°. Axial sites are occupied by O7, O12 of 2 coordinated water with bond angles 78.29 (9)–101.63 (4)°. Geometry around Co(II) ion is distorted elongated Oh with short bonds at equatorial positions at range [1.91 (6)–1.97 (5) Å]. Long bonds formed at axial positions [Co(1)–O(7) is 2.31 (4) Å; Co(1)–O(12) is 2.29 (4) Å]. The axial angle O7–Co1–O6 is 179.09 (4)°. The equatorial angles [O3-Co1-O16; N6-Co1-N14 are 172.34 (2)° and 177.00 (3)°, respectively, are close to linearity. Average Co–O and Co–N bond lengths confirmed Co(II) distorted Oh geometry. Doped Ag^0^NPs had unchanged crystal geometry around the Co(II) ion type, but changed dimensions of the crystal size by expanding the crystal size [35]. The geometry around the Co(II) ion has the same chelating manner, bond length and bond angle as PCC formed distorted Oh, Ag^0^NPs bind Co and proline via Van der Waals interaction [36,37]. All bond lengths and bond angle° of PCC are collected in Appendix A.

The room temperature polycrystalline X-band ESR spectral patterns Figure 11 showed a typical distorted axial pattern for high-spin Co(II) with a three resonances structure. The perpendicular signal is split into two components due to rhombic distortion. Effective g values are g_y_ 2.15, g_x_ 1.98, and g_z_ 1.89 for PCC and g_y_ 2.12, g_x_ 1.98 and g_z_ 1.88 for PCCAg^0^NPs. Hyperfine coupling of electron spin with ^59^Co nucleus (I_3/2_) produces three equally spaced lines in the g_y_ region with [A_y_ 35 × 10^−4^ cm^−1^] and in the g_z_ region with [A_z_ 45 × 10^−4^ cm^−1^] corresponding to |−3/2> → |−1/2>, |+3/2> → |+1/2> and |−1/2> → |−1/2>. Splitting in the g_x_ region is unobserved due to line-width. A_x_ equals 15 × 10^−4^ cm^−1^] [38,39,40].

Molecular orbital calculations, performed by qualitative analysis performed by Ab initio calculations following CASSCF method, Gaussian software program suggested distorted Oh symmetry around Co(II) ion. Elongated Co-O bonds with water molecules in comparison to proline favors magnetic anisotropy, (Figure 12) zero field splitting (ZFS) parameters D and E in Griffith Hamiltonian are obtained using five magnetic parameters: g_x_, g_y_, g_z_, D, and η following Equations (7) and (8), Bond length, Å and bond angle° of PCC are collected in Appendix A [41]:(2)$$D=\frac{3}{2}\;D_{\mathrm{zz}}$$
(3)$$E=\frac{D_{\mathrm{xx}}-D_{\mathrm{yy}}}{2}$$
where parameters D_xx_, D_yy_, and D_zz_ are principal values of ZFS tensor.

ZFS term causes splitting quartet ground state into two Kramers doublets with energy gap, Δ.
(4)$$\Delta =2D\sqrt{1+3\eta^{2}}$$
where η = E/D is the rhombicity parameter. Table 7 showed a positive D value and relative high energy. Splitting between two Kramers doublets nevertheless remains larger than 130 cm^−1^, a value that is within the characteristic range observed for other distorted Oh symmetry. Slow relaxation of single-molecule magnets (SMMs) arises from strong magnetic anisotropy, see Figure 12 and Table 7 and Table 8 [42].

Excitation energy, Δ, includes spin–orbit coupling effects [43]. 

Electrochemical behavior of crystals is represented in Figure 13.

The AC conductivity of the sample is decreased with increasing frequency less than 10^5^ Hz due to charges entrapped between grain boundaries and grains. The AC conductivity of crystals is enhanced by SNPs doping. High electric conductivity of samples enabled the applications as new thermoelectric materials for decreasing heat wastes for the 4th generation of solar cells, new high-temperature superconductors and mini-magnets [44].

Proline showed no antimicrobial activity to any tested species. SNPs enhanced antibacterial activity of Co(II)-proline single crystals based on inhibition zones (IZ, mm), Table 9.

Proline exhibited no inhibition effect on all tested microorganisms. PCC and PCCAg^0^NPs exhibited potent antibacterial and antifungal activities. High microbial activity of PCC crystals is due to dopant Ag^0^NPs. PCC and PCC Ag^0^NPs showed significant inhibition toward *K. pneumoniae*, *E. faecalis*, *A. brasiliensis* and *C. Albicans* in comparison to the reference standard ciprofloxacin antibiotic. Doping by Ag^0^NPs increased chelation complexation between the crystal and DNA of microbes [44].

The concentration response profiles of crystals against MCF-7 and MRC-5 are given in Figure 14a,b and Table 10. PCC and PCCAg^0^NPs exhibited good antitumor activity against human breast adenocarcinoma (MCF-7), IC_50_ values < 62.2 μM. PCCAg^0^NPs exhibited 2-fold more cytotoxicity against MCF-7 (IC_50_ 22.1 μM) than the reference cisplatin, IC_50_ 11.7 μM. Additionally, PCCAg^0^NPs showed 6-fold more cytotoxicity against MCF-7 than PCC (IC_50_ 62.2 μM).

Higher values are observed for IC_50_ of the tested samples for MRC-5 normal healthy lung cells: cisplatin (30.2) < PCCAg^0^NPs (145.5) < PCC (255.8). This trend showed that Ag^0^NPs improved antitumor activity of PCC crystal with low toxicity for normal lung cells.

Same mechanism of action leading antimicrobial and anticancer activity, metal complexes binding DNA of cancer cells.

High cytotoxic activity of PCCAg^0^NPs is due to Ag^0^NPs dopant increased DNA complexation. Cytotoxicity against the breast carcinoma cell line MCF-7 followed the order: PCCAg^0^NPs > PCC > proline.

PCCAg^0^NPs exhibited over 5-fold less cytotoxicity against MRC-5 (IC_50_ 145.5 μM) than the reference cisplatin antitumor therapeutic drug. High toxicity of PCCAg^0^NPs against MRC-5 can be mitigated by using PCCAg^0^NPs in medication as a vial under the applied magnetic field to target tumor cells without affecting normal cells [45].

The dependence of cells viability, human breast adenocarcinoma (MCF-7) and human lung fibroblasts (healthy control) MRC-5, on crystal concentration confirmed that the SNPs crystal showed the best performance in terms of: highest toxicity to cancer cells and less toxicity on the normal lung cell lines approach effect of cisplatin [46,47,48]. All findings obtained in this current study confirmed that silver nanoparticles possess unique physicochemical characteristic-enabled applications in all field technologies [49,50].

## 4. Conclusions

New single crystals of proline amino acid doped by either cobalt chloride (CoCl_2_) or CoCl_2_ + AgNPs are prepared by an innovative low-cost approach (slow evaporation method at room temperature). AgNPs: successfully incorporated into proline single crystals; improved magnetic properties of showed good magnetic properties (µ_eff._ (B.M) for PCC from 4.44 B.M to 4.55 B.M.

Powder XRD diffraction patterns for PCC and PCCAg^0^NPs are obtained in good purity as a triclinic unit cell. Cobalt chloride created anisotropy magnetism in proline single crystals. PCCAg^0^NPs exhibited good antitumor activity against human breast adenocarcinoma (MCF-7). PCCAg^0^NPs crystal showed six-fold more cytotoxicity against breast cancer cell MCF-7, IC_50_ 22.1 μM, than the reference drug cisplatin, IC_50_ 11.7 μM. The mechanism of action of PCCAg0NPs could be binding and complexing DNA of cancer cells. 

PCCAg^0^NPs showed five-fold lower cytotoxicity to normal healthy lung cells MRC-5, (IC_50_ 145.5 μM) than cisplatin (IC_50_ 30.2 μM). The high toxicity of PCCAg^0^NPs against MCF-7 and its low toxicity against MRC-5 makes it a promising candidate for chemotherapy of breast cancer.

PCCAg^0^NPs single crystal showed potent antimicrobial activities to various bacterial and fungal species, inhibition zone (mm): 21, 25, 24, 26, 30, 28, 12, and 46 for *S. aureus*, *E. faecalis*, *S. typhi*, *E. coli*, *P. aerugino*, *K. pneumoniae*, *A. braselienses*, and *C. albicans*, respectively, in comparison to ciprofloxacin (23, 0, 26, 26, 25, 0, 0, 0) for the same tested species.

## Data Availability

All data in this study will be available on request.

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
