# Peer review of "Preparation, Characterization of New Antimicrobial Antitumor Hybrid Semi-Organic Single Crystals of Proline Amino Acid Doped by Silver Nanoparticles"

_biomedicines, 2023, doi:10.3390/biomedicines11020360_

Round 1

Reviewer 1 Report

The revision has been carried out in the attached pdf file.

Author Response

Thank you, Please see the attachment

Thank you for considering my manuscript biomedicines-2039266 entitled (Preparation, characterization of new antimicrobial antitumor hybrid semi-organic single crystals of proline amino acid doped by silver nanoparticles). I would like to inform you that had respond to all reviewers' comments point by point and all changes are highlighted in yellow color.

Comment

Response

Page

Lines

Reviewer 1

1. Of in title should be small letter

Of  replaced by of

Title

2.In abstract, add small introduction first

1

12-14

3.Add reference at (…..electron-hole pair)

Ref.5. is added

2

46

Hybrid semi organic single crystals 4.Abbreviate

HSOSCs

2

67

5.…… are collected in Table1.

Space is left before Table 1

3

81

6.… temperature 25±0.1oC ( unit symbol is not correct)

It is corrected to 25oC ±0.1

3

87

7.Divide experimental section into two parts

It is divided into sections II.1, II.2.

2

78

8.Correct name of bacterial species in Table 8

All names are corrected

23

380

9.Correct Table 8

Table 8 is  corrected

23

24

396-397

411, 413

10.Rewrite conclusion

Conclusion is clarified

25

418-437

Reviewer 2

1. Page 15, figure 8 legend, crystal structure of PCC should be referred to PCCAgNPs. This should be corrected.

Legend of Fig.8 is corrected

18

311

2. Page 20, figure 14, should be placed before table 8. This way the readers are introduced with cytotoxicity first, which also could be verified in table 8. 

Fig.14 is placed before Table 8

24

407-409

3. Conclusion: This section is too short and is only consisting of two paragraphs. It should be expanded by insertion of all data presented in the manuscript. That will enhance the quality of the manuscript and reinforce the specific aims of paper. Specifically, the last sentence should mention IC50 antitumor activity and antibacterial activity as well.

Conclusion is expanded to include all key points obtained in this current study 

25

418-437

Reviewer 3

This paper is interesting and useful for further medical applications. Accordingly, this reviewer recommends publication.

-

-

Reviewer 2 Report

This article is covering some aspects of the Preparation & characterization single crystals of proline & silver nanoparticles. This synthesized sliver nanoparticles of proline have potent antitumor activity against breast cancer MCF-7 and MRC-5 with IC50 190.6 and 295.8 respectively. Additionally, solver complex has antibacterial activity against gram (+) and gram (-) pathogens and antifungal activities as well.

FTIR, ESR spectra and AC conductivity exclusively direct the specific aims of this article on characterization of this new silver complex. The authors proved the presence of single crystal of PCCAgNPs by x-ray crystallography. This will constitute the important goals and novelty of this paper. 

            The following suggested changes and recommendations should be introduced before the publication of the manuscript.

1.     Page 15, figure 8 legend, crystal structure of PCC should be referred to PCCAgNPs. This should be corrected.

2.     Page 20, figure 14, should be placed before table 8. This way the readers are introduced with cytotoxicity first, which also could be verified in table 8. 

3.      Conclusion: This section is too short and is only consisting of two paragraphs. It should be expanded by insertion of all data presented in the manuscript. That will enhance the quality of the manuscript and reinforce the specific aims of paper. Specifically, the last sentence should mention IC50 antitumor activity and antibacterial activity as well.

 The manuscript is of good quality and importance and is sequentially written and edited in order to meet the standard for the articles published inBiomedicines. Thus, I certainly recommend it for publication after the correction of these suggested minor changes and recommendations. 

Author Response

Thank you for considering my manuscript biomedicines-2039266 entitled (Preparation, characterization of new antimicrobial antitumor hybrid semi-organic single crystals of proline amino acid doped by silver nanoparticles). I would like to inform you that had respond to all reviewers' comments point by point and all changes are highlighted in yellow color.

Comment

Response

Page

Lines

Reviewer 1

1. Of in title should be small letter

Of  replaced by of

Title

2.In abstract, add small introduction first

1

12-14

3.Add reference at (…..electron-hole pair)

Ref.5. is added

2

46

Hybrid semi organic single crystals 4.Abbreviate

HSOSCs

2

67

5.…… are collected in Table1.

Space is left before Table 1

3

81

6.… temperature 25±0.1oC ( unit symbol is not correct)

It is corrected to 25oC ±0.1

3

87

7.Divide experimental section into two parts

It is divided into sections II.1, II.2.

2

78

8.Correct name of bacterial species in Table 8

All names are corrected

23

380

9.Correct Table 8

Table 8 is  corrected

23

24

396-397

411, 413

10.Rewrite conclusion

Conclusion is clarified

25

418-437

Reviewer 2

1. Page 15, figure 8 legend, crystal structure of PCC should be referred to PCCAgNPs. This should be corrected.

Legend of Fig.8 is corrected

18

311

2. Page 20, figure 14, should be placed before table 8. This way the readers are introduced with cytotoxicity first, which also could be verified in table 8. 

Fig.14 is placed before Table 8

24

407-409

3. Conclusion: This section is too short and is only consisting of two paragraphs. It should be expanded by insertion of all data presented in the manuscript. That will enhance the quality of the manuscript and reinforce the specific aims of paper. Specifically, the last sentence should mention IC50 antitumor activity and antibacterial activity as well.

Conclusion is expanded to include all key points obtained in this current study 

25

418-437

Reviewer 3

This paper is interesting and useful for further medical applications. Accordingly, this reviewer recommends publication.

-

-

Reviewer 3 Report

In the manuscript “Preparation, Characterization Of New Antimicrobial Antitumor Hybrid Semi-Organic Single Crystals Of Proline Amino Acid Doped By Silver Nanoparticles”, Ali et al prepared and characterized four new colored single crystals of proline doped by cobalt chloride in absence and presence of doping SNPs. The properties of these crystals were also compared to those of proline and glycine single crystals. Crystals are successfully grown by slow evaporation method at room temperature. In addition, the crystals doped by SNPs possessed potent antitumor activity. This paper is interesting and useful for further medical applications. Accordingly, this reviewer recommends publication.

Author Response

(The authors gave the same response as above.)
